# Developmental Exposure to DDT Disrupts Transcriptional Regulation of Postnatal Growth and Cell Renewal of Adrenal Medulla

**DOI:** 10.3390/ijms24032774

**Published:** 2023-02-01

**Authors:** Nataliya V. Yaglova, Svetlana V. Nazimova, Sergey S. Obernikhin, Dibakhan A. Tsomartova, Valentin V. Yaglov, Ekaterina P. Timokhina, Elina S. Tsomartova, Elizaveta V. Chereshneva, Marina Y. Ivanova, Tatiana A. Lomanovskaya

**Affiliations:** 1Laboratory of Endocrine System Development, A.P. Avtsyn Research Institute of Human Morphology of Federal State Budgetary Scientific Institution “Petrovsky National Research Center of Surgery”, 119991 Moscow, Russia; 2Department of Human Anatomy and Histology, Federal State Funded Educational Institution of Higher Education I.M. Sechenov First Moscow State Medical University, 119435 Moscow, Russia

**Keywords:** adrenal medulla, DDT, endocrine-disrupting chemicals, growth, self-renewal, Oct4, Shh

## Abstract

Dichlorodiphenyltrichloroethane (DDT) is the most widespread persistent pollutant with endocrine-disrupting properties. DDT has been shown to disrupt secretory and morphogenetic processes in the adrenal cortex. The present investigation aimed to evaluate transcriptional regulation of postnatal growth of the adrenal medulla and formation of the pools necessary for self-renewal of medullary cells in rats that developed under low-dose exposure to DDT. The study was performed using male Wistar rats exposed to low doses of o,p’-DDT during prenatal and postnatal development. Light microscopy and histomorphometry revealed diminished medulla growth in the DDT-exposed rats. Evaluation of Ki-67 expression in chromaffin cells found later activation of proliferation indicative of retarded growth of the adrenal medulla. All DDT-exposed rats exhibited a gradual decrease in tyrosine hydroxylase production by adrenal chromaffin cells. Immunohistochemical evaluation of nuclear β-catenin, transcription factor Oct4, and ligand of sonic hedgehog revealed increased expression of all factors after termination of growth in the control rats. The DDT-exposed rats demonstrated diminished increases in Oct4 and sonic hedgehog expression and lower levels of canonical Wnt signaling activation. Thus, developmental exposure to the endocrine disruptor o,p’-DDT alters the transcriptional regulation of morphogenetic processes in the adrenal medulla and evokes a slowdown in its growth and in the formation of a reserve pool of cells capable of dedifferentiation and proliferation that maintain cellular homeostasis in adult adrenals.

## 1. Introduction

Negative outcomes of exposure to endocrine-disrupting chemicals are commonly associated with impaired activation of hormonal receptors or secretory disorders in endocrine cells [1,2]. However, a growing body of evidence indicates that endocrine disruptors may interfere with cell, tissue, and organ fate during prenatal and postnatal development, particularly, but not only, of the endocrine and immune systems [3,4,5,6]. Since endogenous hormones regulate the fate of all cells in the body (proliferation, apoptosis, differentiation, and migration), a violation of hormone signaling can cause disruption to the implementation of the genetic program for organ development, both in the embryo and after birth. Changes in the program of development and functional maturation of organs are known to arise from impaired morphogenetic processes during postnatal ontogeny, specifically from a misbalance between the proliferation and differentiation of parenchyma cells. Dysmorphogenetic properties have already been found in a wide range of endocrine-disrupting chemicals, including polychlorinated biphenyls, oxybenzone, phthalates, and some organochlorine pesticides [7,8,9,10,11].

Dichlorodiphenyltrichloroethane (DDT) is the most widespread persistent organic pollutant with endocrine-disrupting properties on the planet [12,13,14]. DDT is still used for fighting vector-prone diseases such as malaria, leishmaniasis, dengue fever, and others [14]. The extensive use of DDT in the twentieth century and its continued use today, as well as its long half-life, contribute to the spread and persistence of this pesticide in the environment. Investigations have shown that even low-dose exposure to DDT affects both male and female reproduction systems and disrupts thyroid and adrenal function [1,12,13,14,15,16,17,18,19]. Analysis of scientific reports and the results of our previous investigations suggest consideration of DDT as a dysmorphogenetic factor as well as a disruptor of secretory function and receptor-ligand interactions. DDT easily penetrates the placental barrier due to its high lipophilicity and small molecular weight [20]. It exerts a direct influence on the endocrine disruptor in both maternal and fetal tissues. A striking example of the disruption of fetal organ formation by DDT is the malformation of male genitalia reported in numerous publications [21,22,23,24,25]. The dysmorphogenetic effect of DDT on endocrine glands has been less studied. Our previous investigations have revealed significant changes in the morphogenesis of the adrenal cortex after prenatal and postnatal exposure to low doses of DDT [26,27,28]. Low-dose exposure has also been found to affect adrenal medulla function [18]. In our previous studies, we showed that decreased epinephrine production is associated with insufficiency of the secretory machinery of chromaffin cells [29]. It is possible that the impaired age-related rearrangement of the secretory apparatus of chromaffin cells reflects changes in postnatal morphogenesis, which requires further research. We aimed to study the transcriptional regulation of postnatal growth of the adrenal medulla and the formation of the pools necessary for the self-renewal of medullary cells in rats that developed under low-dose exposure to DDT during prenatal and postnatal periods of ontogeny.

## 2. Results

### 2.1. Adrenal Medulla Histology

The adrenal medulla of the pubertal 6-week-old control rats was composed of clustered chromaffin cells separated by thin connective tissue septa and abundant capillaries and venous sinusoids. The chromaffin cells had large round nuclei and slightly oxyphilic cytoplasm with basophilic inclusions (Figure 1A). Single neurons were also observed in the adrenal medulla sections. 

The 6-week-old pubertal rats exposed prenatally and postnatally to DDT showed no significant differences in adrenal medulla histology (Figure 1B). The surface area of the adrenal medulla and total area of chromaffin cells were within the control ranges (Figure 1E,F). The adrenal medulla was composed of a larger number of chromaffin cells, which were smaller in size than the control values (Figure 1G,H). 

After sexual maturation by 10 weeks of age, the medullary cells of the control rats exhibited more basophilic cytoplasm compared to the previous age (Figure 1C). The adrenal medulla increased in size by 20% (Figure 1E). The total area of chromaffin cells in the adrenal sections proportionally increased compared to the previous age, as confirmed by determining the area of chromaffin cells and the number of cells in the sections, which increased by 25%. Enlargement of the adrenal medulla was clearly associated with the increase in number, not in size, of the chromaffin cells, as shown in Figure 1F–H. 

After puberty, the chromaffin cells in the DDT-exposed rats demonstrated enlightened and less basophilic cytoplasm (Figure 1D). The surface area of the adrenal medulla was 25% smaller than in the controls. The total area and number of chromaffin cells were also diminished. The size of the chromaffin cells did not significantly change with age and was within the control range (Figure 1E–H). Thus, light microscopy and histomorphometry did not reveal growth of the adrenal medulla in the DDT-exposed rats.

### 2.2. Proliferation of Chromaffin Cells

Evaluation of Ki-67 expression found that Ki-67-positive chromaffin cells reduced with age in the control rats (Figure 2). The developmentally DDT-exposed rats showed a significantly smaller percentage of Ki-67-positive chromaffin cells at the age of 6 weeks and a significant rise in proliferative activity by the age of 10 weeks (Figure 2). 

### 2.3. Tyrosine Hydroxylase Content in Chromaffin Cells

All of the chromaffin cells of the control rats exhibited high tyrosine hydroxylase content in the cytoplasm at both pubertal and postpubertal ages (Figure 3A,B). The developmentally exposed pubertal rats demonstrated lower tyrosine hydroxylase content at both ages. Light microscopy also revealed populations of chromaffin cells with extremely low tyrosine hydroxylase content, which were absent in the control rats (Figure 3C–E). The percentage of chromaffin cells with low-to-negative tyrosine hydroxylase content was found to increase significantly with age in the exposed rats (Figure 3E).

### 2.4. Activation of Canonical Wnt Signaling in Adrenal Chromaffin Cells

β-catenin was observed mainly in the outer membranes and nuclei of the chromaffin cells in the controls. Chromaffin cells with nuclear localization of β-catenin, indicative of canonical Wnt signaling activation, were observed in the adrenal medulla of the pubertal and postpubertal control rats (Figure 4A,B). The control rats showed a three-fold increase in the percentage of chromaffin cells with β-catenin-positive nuclei after puberty (Figure 4E).

In 6-week-old rats exposed to DDT during prenatal and postnatal development, less β-catenin was produced by the chromaffin cells (Figure 4C). The percentage of cells with β-catenin-positive nuclei was almost half of the control values (Figure 4E). After puberty, the percentage of cells with β-catenin translocated into the nucleus increased 3.15 times, but was 1.5 times less than the values of the control group (Figure 4D,E).

### 2.5. Sonic Hedgehog (Shh) Expression by Adrenal Chromaffin Cells

In the pubertal control rats, Shh-positive cells were extremely rare in the medulla. However, after puberty, their number increased drastically (Figure 5A,B,E). Shh ligands were observed only in the nuclei of the chromaffin cells. 

In the rats that were developmentally exposed to low doses of DDT, the percentage of Shh-positive cells exceeded the control values at puberty. In contrast to the control animals, the percentage increase of Shh-positive cells with age was significantly lower (Figure 5C–E).

### 2.6. Oct4 Expression by Adrenal Chromaffin Cells

Immunohistochemical evaluation of the adrenal medulla sections revealed Oct4-positive cells in the pubertal control rats. The Oct4-producing cells exhibited were typical for chromaffin cell morphology, with nuclear localization of Oct4 (Figure 6A). After puberty, their number tripled (Figure 6B,E).

The percentage of Oct4-positive cells in the developmentally DDT-exposed rats of pubertal age did not differ from the controls (Figure 6C,E). All the immunopositive cells demonstrated nuclear localization of Oct4. After puberty, their number increased but to a lesser extent than in the control group (Figure 6D,E).

## 3. Discussion

The data obtained show that the growth of the adrenal medulla in rats prenatally and postnatally exposed to the endocrine disruptor DDT differs from the control parameters. The main mechanism for the growth of inner organs is the proliferation of parenchymal and stromal cells. In intact rats, we observed an increase in the number of chromaffin cells and the corresponding development of the microcirculatory bed. After reaching maximal development, the proliferative activity of the cells decreased. In rats developmentally exposed to DDT, there was a smaller increase in the number of chromaffin cells, and, subsequently, there was no increase in the size of the medulla. In contrast to the intact rats, the chromaffin cells increased in size instead of increasing in number. The rise in proliferative activity found in postpubertal rats indicates the activation of growth. Therefore, in rats exposed to DDT, there was not a premature arrest but a slowdown in the growth rates of the adrenal medulla. An important difference was the reduced content of tyrosine hydroxylase in the cytoplasm of the chromaffin cells. Thyrosine hydroxylase is a recognized marker of chromaffin cells differentiation [30]. The observed gradual increase in the percentage of cells with an extremely low level of tyrosine hydroxylase indicates either inhibition of its synthesis by the disruptor or an increase in low-differentiated cells, which might result from altered transcriptional control of cell proliferation and differentiation.

In the intact rats, the termination of adrenal growth was associated with a rise in the activation of Wnt signaling. Canonical β-catenin/Wnt signaling is known as a key pathway that regulates cell proliferation, migration, differentiation, and survival [31]. Wnt signaling also mediates self-renewal signals in various cell types [32,33,34]. The implications of Wnt signaling in the prenatal and postnatal development of chromaffin tissue in adrenals are still poorly studied. Single reports demonstrate that the inactivation of the β-catenin gene in neural crest cells results in malformations of the midbrain and hindbrain, reduction of sensory cells and melanocytes, and loss of adrenal chromaffin cell progenitors [35,36,37]. Wnt signaling is known to promote the growth of the zona glomerulosa and zona reticularis and to inhibit the maturation of fasciculata cells [38,39,40]. Unlike adrenocortical cells, which maintain constant levels of canonical Wnt signaling [41], chromaffin cells have been found to enhance the translocation of β-catenin into nuclei with age. β-catenin-dependent Wnt pathways have been shown to play a pivotal role in the expansion and progression of neural tumors [35]. These facts suggest the implication of Wnt signaling in the formation of a pool of chromaffin cells for self-renewal. The DDT-exposed rats demonstrated the same dynamics of Wnt signaling activation, but at a lower range. The inhibited translocation of β-catenin into nuclei was probably associated with the slowdown in adrenal medulla growth and, consequently, with the delayed formation of the niche for the self-renewal of chromaffin cells after growth termination.

Oct4, a stemness marker, is also known as a transcription factor capable of reprogramming somatic cells and an integrator of pluripotency and self-induced differentiation [42,43,44]. Adrenal chromaffin cells have already been shown to express Oct4 during postnatal development, and the population of Oct4-positive chromaffin cells increases in number after puberty [45,46,47]. Our investigation revealed that all Oct4-positive cells in the intact rats had typical chromaffin cell morphology. Since all the chromaffin cells of the intact rats actively produced tyrosine hydroxylase, the Oct4-positive cells also expressed it. This higher percentage of typical Oct4-coexpressing chromaffin cells allows us to consider it as a mechanism for cell maintenance by dedifferentiation and proliferation. Therefore, the slowdown in the formation of the Oct4-positive cell pool after puberty in DDT-exposed rats could be related to the slow growth of the adrenal medulla.

The Shh pathway is an essential factor in the regulation of tissue homeostasis and repair [48]. Shh is a morphogen expressed in neurons and glial cells and is considered a pathway providing recruitment of distinct cell populations and generating new cells for repair in the central nervous system [49,50,51,52]. Regeneration of the adrenal cortex also requires Shh signaling [53]. Our findings are consistent with the above-mentioned reports. Shh expression in the chromaffin cells of the intact rats demonstrated an increase after the termination of adrenal growth as well as Oct4 induction. It is noteworthy that all of the Shh-positive cells exhibited phenotype of typical chromaffin cells actively expressing tyrosine hydroxylase. In the rats exposed to the endocrine disruptor, the slowdown in the growth of the adrenal medulla was accompanied by a lower formation of a Shh-positive pool of cells. In contrast to Oct4, the percentage of Shh-positive cells in the pubertal period exceeded the control values. It is likely that this surge was associated with signals of the arrest of adrenal medulla growth, and further activation of the chromaffin cell proliferation inhibited the formation of the Shh-positive cell pool.

The results of the study show that the activation of morphogens and pluripotency factors in the chromaffin cells of intact rats occur in parallel with the high expression of tyrosine hydroxylase, a terminal differentiation marker for chromaffin cells. Therefore, the decrease in tyrosine hydroxylase expression in animals exposed to DDT seems not to be associated with changes in morphogenesis. It is quite possible that tyrosine hydroxylase synthesis is directly inhibited by DDT. This suggestion requires additional investigation. The downregulation of tyrosine hydroxylase expression is a negative factor since it may result in insufficient catecholamine production in response to stress. The present results help shed light on previous data showing impaired mitochondrial rearrangement [29]. It is likely that the apparent lag in adrenal medulla development and its transcriptional regulation also explains the impaired age-related rearrangement of the secretory apparatus of the cells. 

## 4. Materials and Methods

### 4.1. Animals

Female and male Wistar rats of the same age were obtained from the Scientific Center of Biomedical Technologies at the Federal Medical-Biological Agency of Russia. All rats were housed in a local vivarium with 5 animals per cage. The housing conditions were: temperature +22–23 °C, humidity 60–65%, and light/darkness cycle 12/12 h. The rats were given standard pelleted chow. Access to food and water was free. The investigation was performed in accordance with the handling standards and rules of laboratory animals as consistent with the International Guidelines for Biomedical Research with Animals (1985), routine laboratory standards in the Russian Federation (Order of the Ministry of Healthcare of the Russian Federation No. 267 dated 19 June 2003), the Animal Cruelty Protection Act (dated 1 December 1999), and regulations of experimental animal operation (Order of the Ministry of Healthcare of the USSR No. 577 dated 12 August 1977). Animal procedures were approved by the Ethics Committee at the Research Institute of Human Morphology (protocol 28(4) dated 28 October 2021).

### 4.2. Experimental Design

The female rats weighed 180–220 g and received a solution of o,p-DDT in tap water with concentration 20 µg/L (Sigma-Aldrich, St. Louise, MO, USA) ad libitum since mating during pregnancy and lactation. The o,p-isomer of DDT was preferred due to its higher solubility in water (80 µg/L) [54]. After weaning (3 weeks of age), the progeny of the rat dams received the same solution of o,p-DDT during postnatal development up to 10 weeks of age, when rat adrenals reach their maximal development [55]. No differences in terms of pregnancy, number of pups, or female/male ratio between the controls and the exposed rats were observed. Only male progeny were enrolled for examination (*n* = 20) to avoid changes in some morphofunctional parameters of the adrenals evoked by fluctuations in female sex hormones during the ovarian cycle. Male progeny of intact female rats was used as controls (*n* = 20). Half of the rats were sacrificed in the pubertal period at the age of 6 weeks and the others after puberty at the age of 10 weeks by Zoletil overdose. Body weight and amount of consumed water were measured daily to calculate daily intake of DDT. The exposed and control progeny had similar values of body weight during postnatal development. The average daily intake of DDT by the dams during pregnancy and lactation was 2.69 ± 0.18 µg/kg bw and by progeny after weaning was 2.90 ± 0.12 µg/kg bw, which corresponded to DDT consumption by humans through food products, with consideration for differences in DDT metabolism between rats and humans [56]. No differences were found in the amount of food consumed. The body masses of the 6-week rats were 257.8 ± 8.2 gg for the control rats and 254.4 ± 7.8 gg for the DDT-exposed rats. At the age of 10 weeks, the body masses of the control and DDT-exposed rats were also similar at 345.0 ± 17.4 and 349 ± 15.0, respectively. The absence of DDT, its metabolites, and related organochlorine compounds in the tap water and chow was confirmed by gas chromatography at the Moscow Federal Budgetary Institution of Public Health. 

### 4.3. Adrenal Histology

The adrenal glands were fixed in Bouin solution. After standard histological processing, the tissue samples were embedded in paraffin. Equatorial sections of the adrenals were stained with hematoxylin and eosin. Histological examination was performed using a Leica DM2500 light microscope (Leica Microsystems Gmbh, Wetzlar, Germany). Computer histomorphometry of the light microscope images was carried out using ImageScope software (Leica Microsystems Gmbh, Wetzlar, Germany). The surface area of the adrenal medulla and chromaffin cells and the size and number of chromaffin cells were measured.

### 4.4. Immunohistochemistry

Immunohistochemical evaluation of Ki-67, tyrosine hydroxylase, β-catenin, sonic hedgehog (Shh), and Oct4 was performed on the paraffin-embedded tissues. After antigen retrieval with 10 mM sodium citrate (pH 6.0), endogenous peroxidase and endogenous immunoglobulins were blocked using Hydrogen Peroxide Block and Protein Block (Thermo Fisher Scientific, Waltham, MA, USA). The slides were incubated overnight at 8 °C with primary antibodies for Ki-67 (1:100, Cell Marque, Rocklin, CA, USA), tyrosine hydroxylase (1:1000, Abcam, Cambridge, MA, USA), β-catenin (1:100, Cell Marque, Roklin, CA, USA), Shh (1:400, Abcam, Cambridge, MA, USA), and Oct4 (1:5000, Abcam, Cambridge, MA, USA) overnight at 8 °C. Sections of embryonic rat tissues were used as a positive control for Oct4. Slides with primary antibodies processed without incubation were used as a negative control. The reaction was visualized using an UltraVision LP Detection System reagent kit (Thermo Fisher Scientific, Waltham, MA, USA) according to the manufacturer’s recommendations. The sections were counterstained with Mayer’s hematoxylin.

Activation of canonical Wnt signaling was assessed by the percentage of chromaffin cells with β-catenin-positive nuclei [56,57]. Tyrosine hydroxylase-positive cells were examined at a magnification of 1000 to detect chromaffin cells with cytoplasm containing tyrosine hydroxylase content of less than 10%, which were referred to as low-to-negative cells. The percentages of positive and low-to-negative tyrosine hydroxylase chromaffin cells were calculated. Ki-67, Shh, and Oct4 expression was assessed as percentages of immunopositive cells with nuclear staining. 

### 4.5. Statistical Analysis

The statistical analyses were carried out using the software package Statistica 7.0 (StatSoft, Tulsa, OK, USA). The central tendency and dispersion of quantitative traits with approximately normal distribution were presented as the mean and standard error of the mean (M ± SEM). Quantitative comparisons of independent groups were performed using the Student’s *t*-test, taking into account the values of Levene’s test for the equality of variances. Quantitative comparisons were performed using chi-square tests. Differences were considered statistically significant at *p* < 0.05.

## 5. Conclusions

Low-dose exposure to the endocrine disruptor DDT during prenatal and postnatal development alters the transcriptional regulation of morphogenetic processes in the adrenal medulla of male rats, evoking a slowdown in its growth and in the formation of a reserve pool of cells capable of dedifferentiation and proliferation that maintain cellular homeostasis in adult adrenals. The diminished regeneration capacity of the adrenal medulla may, in turn, negatively affect adrenal response to physiological and pathological factors and disturb adaptation to stress. The present findings indicate the multiplicity of mechanisms of the disruptive action of DDT, which disturbs both the development and the response of the adrenal medulla to stress factors. 

## Figures and Tables

**Figure 1 ijms-24-02774-f001:**
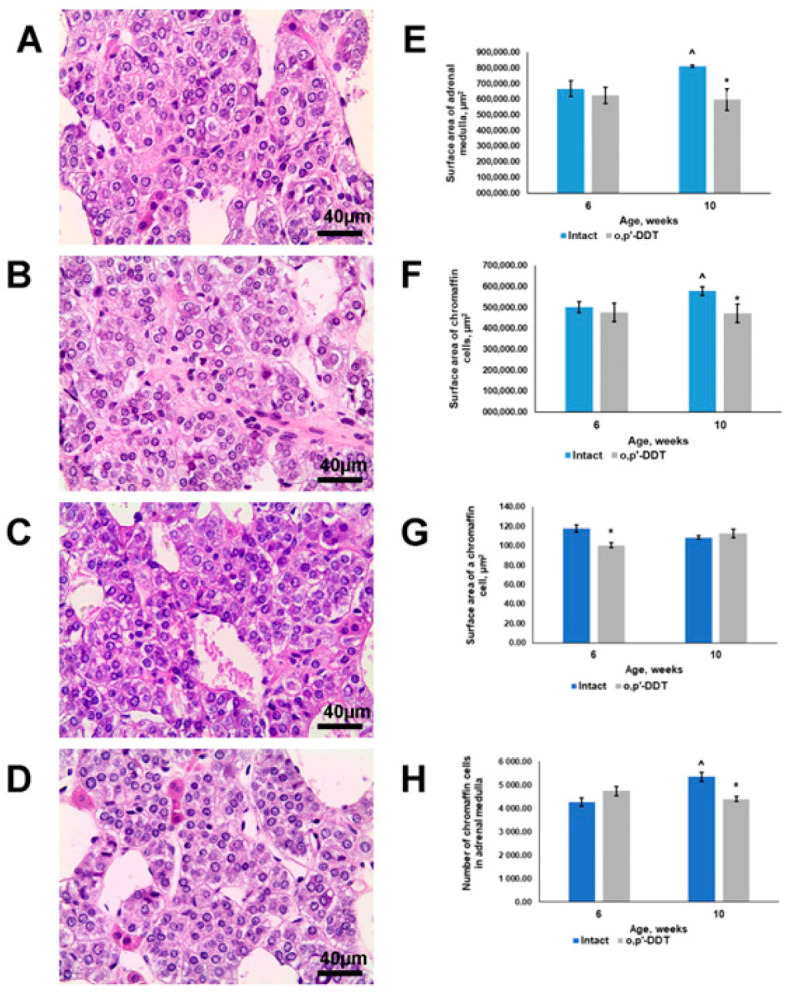
Changes in adrenal medulla histology and histomorphometry in rats developmentally exposed to DDT. Histology of adrenal medulla of control rats (**A**) and DDT-exposed rats (**B**) in pubertal period. Histology of control rats (**C**) and DDT-exposed rats (**D**) after puberty. Magnification 400, scale bar 40 µm. Surface area of the adrenal medulla (**E**), total surface area of chromaffin cells (**F**), surface area of a chromaffin cell (**G**), and number of chromaffin cells in the sections (**H**). Data are shown as mean ± S.E.M. *p* < 0.05 compared to the control (*), compared to 6-week age (^).

**Figure 2 ijms-24-02774-f002:**
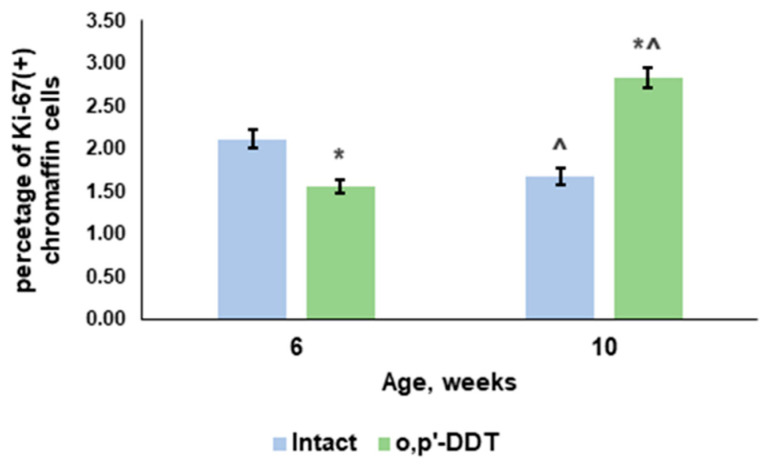
Changes in proliferative activity of adrenal chromaffin cells in rats developmentally exposed to DDT. Data are shown as mean ± S.E.M. *p* < 0.05 compared to the control (*), compared to 6-week age (^).

**Figure 3 ijms-24-02774-f003:**
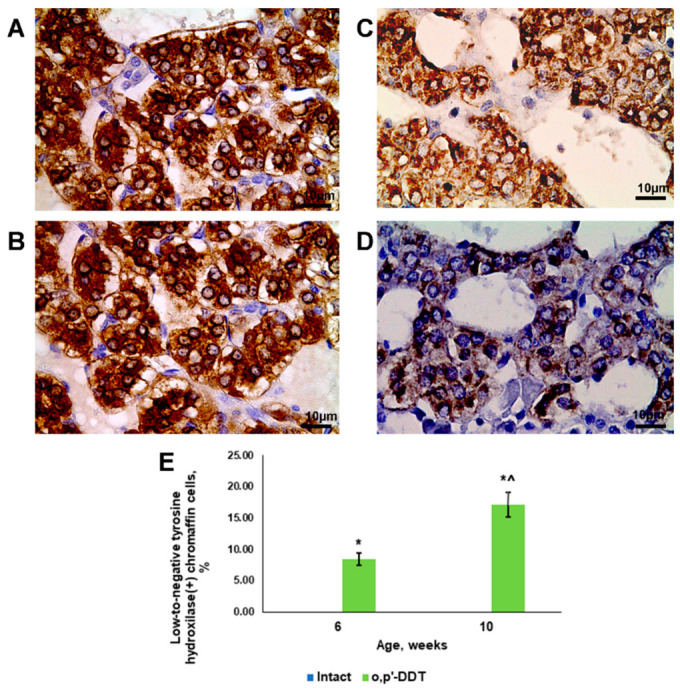
Changes in tyrosine hydroxylase content in adrenal chromaffin cells in rats developmentally exposed to DDT. Immunohistochemical detection of tyrosine hydroxylase in control rats in pubertal (**A**) and postpubertal (**B**) periods and in DDT-exposed rats in pubertal (**C**) and postpubertal (**D**) periods. Magnification 800, scale bar 10 µm. Percentage of low-to-negative tyrosine hydroxylase-expressing chromaffin cells (**E**). Data are shown as mean ± S.E.M. *p* < 0.05 compared to the control (*), compared to 6-week age (^).

**Figure 4 ijms-24-02774-f004:**
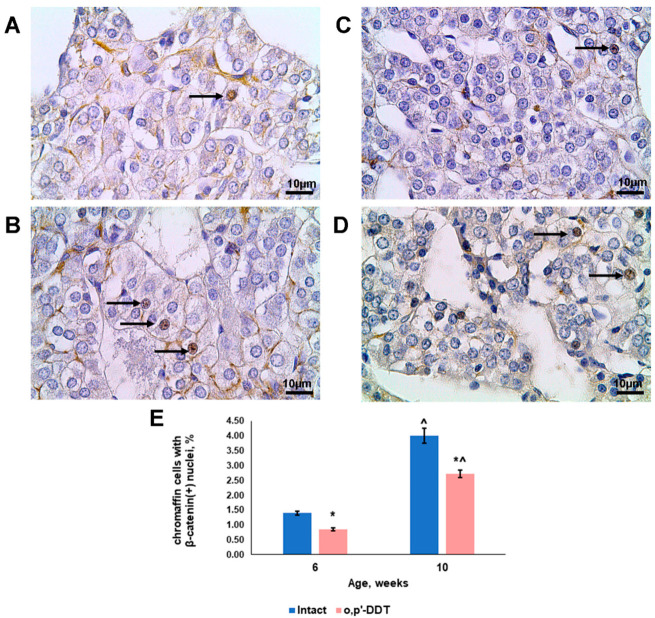
Changes in β-catenin production by adrenal chromaffin cells in rats developmentally exposed to DDT. Immunohistochemical detection of β-catenin in control rats in pubertal (**A**) and postpubertal (**B**) periods and in DDT-exposed rats in pubertal (**C**) and postpubertal (**D**) periods. Magnification 800, scale bar 10 µm. Arrows point to β-catenin-positive cells. Percentage of chromaffin cells with β-catenin-positive nuclei (**E**). Data are shown as mean ± S.E.M. *p* < 0.05 compared to the control (*), compared to 6-week age (^).

**Figure 5 ijms-24-02774-f005:**
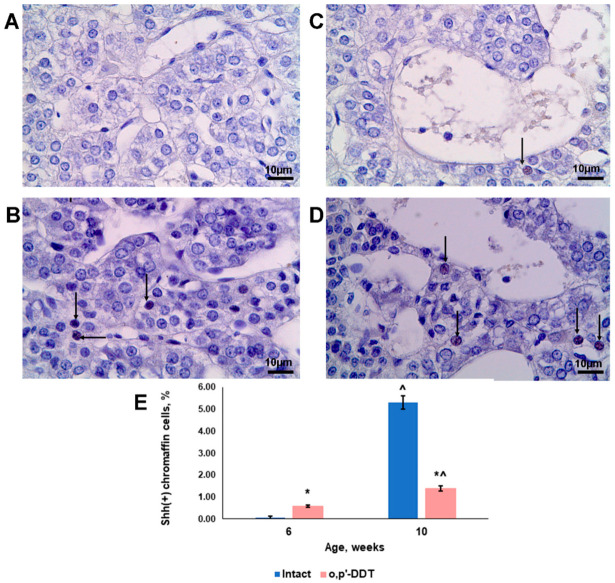
Changes in Shh expression in adrenal chromaffin cells of rats developmentally exposed to DDT. Immunohistochemical detection of Shh in control rats in pubertal (**A**) and postpubertal (**B**) periods and in DDT-exposed rats in pubertal (**C**) and postpubertal (**D**) periods. Magnification 800, scale bar 10 µm. Arrowheads point to Shh-positive cells. Percentage of Shh-positive chromaffin cells (**E**). Data are shown as mean ± S.E.M. *p* < 0.05 compared to the control (*), compared to 6-week age (^).

**Figure 6 ijms-24-02774-f006:**
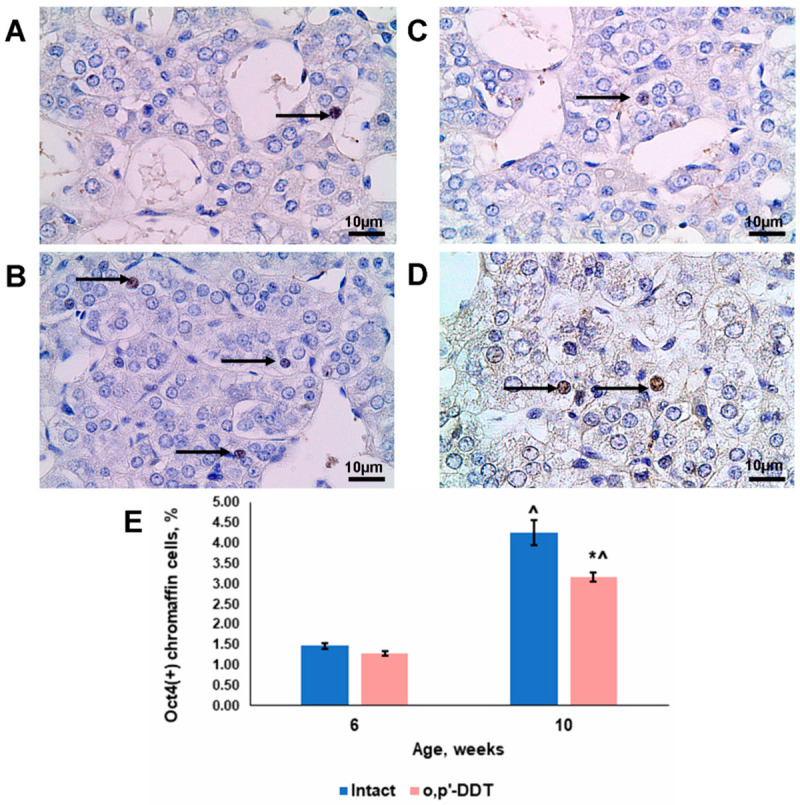
Changes in Oct4 expression in adrenal chromaffin cells of rats developmentally exposed to DDT. Immunohistochemical detection of Oct4 in control rats in pubertal (**A**) and postpubertal (**B**) periods and in DDT-exposed rats in pubertal (**C**) and postpubertal (**D**) periods. Magnification 800, scale bar 10 µm. Arrowheads point to Oct4-positive cells. Percentage of Oct4-positive chromaffin cells (**E**). Data are shown as mean ± S.E.M. *p* < 0.05 compared to the control (*), compared to 6-week age (^).

## Data Availability

The data presented in this study are available from the corresponding author upon reasonable request.

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
