# Peer review of "Developmental Exposure to DDT Disrupts Transcriptional Regulation of Postnatal Growth and Cell Renewal of Adrenal Medulla"

_ijms, 2023, doi:10.3390/ijms24032774_

Round 1

Reviewer 1 Report

The article is well-written and do not require revision. Just add the information why you preferred o,p’-isomer of DDT, not p,p’-one to the Materials and methods.

The dot at the end of the article title is not needed.

The first affiliation has an extra space in the e-mail.

The font size in the labels of the axes in figures 1 and 2 is different, it is necessary to unify.

Author Response

Dear Reviewer! We thank you for your attention to our manuscript. We have made all necessary corrections to the text of the article. 

Reviewer 2 Report

The paper 'Developmental exposure to DDT disrupts transcriptional regulation of postnatal growth and cell renewal of adrenal medulla' provides several interesting data on the effects of DDT on adrenal development. Unfortunately, since the experimental design is extremely poor, the reliability of the data presented is consequently strongly affected.

In particular:

The Authors should provide a clear picture of the experimental design. How many days did the treatment last?

Please describe in detail the housing condition of the rats during the treatment period

It is known that DDT is a lipophilic compound (it has been declared by the Authors): why the administration in tap water? What kind of vehicle has been used to dissolve DDT? Please provide explanation of lines 253-254

How many days did the treatment last in total?

Please specify if the concentration of DDT has been adjusted for the body weight of animals during the treatment period.

Please provide info on mating, on male rats used for mating, the fate of male rats after the mating.

Please provide data on the body weight, body weight gain and feed consumption of female rats and dams.

When the dams delivered? The number/sex/weight of pups ?

The use of female rats as negative control is unacceptable. The Authors should know that several sex differences exist in the response to chemicals, above all considering endocrine disrupters. 

The Authors declare that the average daily intake of DDT after weaning was 2.90±0.12 μg/kg bw: please provide the same data for the female rats since mating and during pregnancy and lactation.

Please provide explanation why only male rats have been used for the experiment: this can be considered a strong bias.

Moreover,

In the Introduction, lines 36, 37, 38: I should say that this sentence declares a consolidated mode of action of EDCs. They interfere with cells and tissues of the endocrine system (not only) and this is the main reason for what they are defined 'endocrine disrupting..'.

Lines 51-52 This sentence has no meaning. Please explain

Results. paragraph 2.3 It is not clear which treatment group the figure #3 is related to. Apparently all the panels belong to control group. No images on treated groups are then provided. It should be better explained

Round 2

Reviewer 2 Report

Response 10: Examination of both sexes is incorrect due to influence of female sex steroids. Some morphofunctional parameters of the adrenals are known to vary depending on phase of ovarian cycle. That is why males are usually used. The explanation is added to the text (lines 269-271).

This comment provided by the Authors is not acceptable. The examination of both sexes is necessary and unavoidable to obtain sound and reliable data. The influence of sex steroid fluctuation in female rats can be managed checking the oestrous cyclicity in all the animals and using suitable control group.

Animal experiment

Umidity? Light/dark cycle? Several info an animal housing conditions are lacking. Please report them on the paper and not in the response to Referee.

Experimental design line 256: what the Authors mean with 'istead of tap water'?

Data on food consumption should be provided together with body weight gain for all the animals involved in the experiment, as basic rule of toxicological study.

Is Bouin solution, not Bouen

The Authors have to clearly identify in the Conclusions such bias, underlying that the data obtained in the present work derive from the examination of only one sex.
